# Combining Analytical Approaches and Multiple Sources of Information to Improve Interpretation of Diagnostic Test Results for Tuberculosis in Wild Meerkats

**DOI:** 10.3390/ani11123453

**Published:** 2021-12-04

**Authors:** Stuart J. Patterson, Charlene Clarke, Tim H. Clutton-Brock, Michele A. Miller, Sven D. C. Parsons, Dirk U. Pfeiffer, Timothée Vergne, Julian A. Drewe

**Affiliations:** 1Veterinary Epidemiology, Economics and Public Health Group, Royal Veterinary College, University of London, Hawkshead Lane, Hatfield AL9 7TA, UK; dirk.pfeiffer@cityu.edu.hk (D.U.P.); jdrewe@rvc.ac.uk (J.A.D.); 2SAMRC Centre for TB Research, DSI/NRF Centre of Excellence for Biomedical Tuberculosis Research, Division of Molecular Biology and Human Genetics, Faculty of Medicine and Health Sciences, Stellenbosch University, Cape Town 8000, South Africa; clarke.charlene0303@gmail.com (C.C.); miller@sun.ac.za (M.A.M.); svenparsons@hotmail.com (S.D.C.P.); 3Large Animal Research Group, Department of Zoology, University of Cambridge, Downing Street, Cambridge CB2 3EJ, UK; thcb@cam.ac.uk; 4Mammal Research Institute, University of Pretoria, Hatfield, Pretoria 0028, South Africa; 5Department of Infectious Diseases and Public Health, Jockey Club College of Veterinary Medicine and Life Sciences, City University of Hong Kong, Hong Kong, China; 6UMR ENVT-INRAE IHAP, National Veterinary School of Toulouse, 31300 Toulous, France; timothee.vergne@envt.fr

**Keywords:** diagnostics, interpretation, wildlife, tuberculosis, meerkats

## Abstract

**Simple Summary:**

Diagnostic tests used in animals are ideally extensively trialled to ensure that practitioners have confidence in the results; the ideal target should be 100% of infected animals testing positive (sensitivity), and 100% of uninfected animals testing negative (specificity). In these trials, a gold standard is necessary, against which the diagnostic tests may be compared. Commonly, tests of wild animals are not extensively trialled due to shortages of samples and the rarity of gold standard tests. This study uses samples collected for the purpose of diagnosing *Mycobacterium suricattae* infection in meerkats and estimates the sensitivity and specificity of available tests, both individually, and in combination. A small number of post-mortems (gold standard) were carried out, providing definitive evidence of infected animals against which to compare the tests. In addition, test results were unconventionally compared to survival times and clinical characteristics, aiming to quantify the prognostic capabilities of the tests. The study therefore not only estimates the required parameters against which to assess these tests, but also provides a model for future assessment of diagnostic tests in imperfect field scenarios. Wildlife diseases are increasingly recognised as important to society, and so methods to better quantify these are increasingly important.

**Abstract:**

Diagnostic tests are used to classify individual animals’ infection statuses. However, validating test performance in wild animals without gold standard tests is extremely challenging, and the issue is further complicated in chronic conditions where measured immune parameters vary over time. Here, we demonstrate the value of combining evidence from different diagnostic approaches to aid interpretation in the absence of gold standards, large sample sizes, and controlled environments. Over a two-year period, we sampled 268 free-living meerkats (*Suricata suricatta*) longitudinally for *Mycobacterium suricattae* (a causative agent of tuberculosis), using three ante-mortem diagnostic tests based on mycobacterial culture, and antigen-specific humoral and cell-mediated immune responses, interpreting results both independently and in combination. Post-mortem cultures confirmed *M. suricattae* infection in 22 animals, which had prior ante-mortem information, 59% (13/22) of which were test-positive on a parallel test interpretation (PTI) of the three ante-mortem diagnostic assays (95% confidence interval: 37–79%). A similar ability to detect infection, 65.7% (95% credible interval: 42.7–84.7%), was estimated using a Bayesian approach to examine PTI. Strong evidence was found for a near doubling of the hazard of death (Hazard Ratio 1.75, CI: 1.14–2.67, *p* = 0.01), associated with a positive PTI result, thus demonstrating that these test results are related to disease outcomes. For individual tests, small sample sizes led to wide confidence intervals, but replication of conclusions, using different methods, increased our confidence in these results. This study demonstrates that combining multiple methodologies to evaluate diagnostic tests in free-ranging wildlife populations can be a useful approach for exploiting such valuable datasets.

## 1. Introduction

In order to have confidence in the results of a diagnostic test, the user must have knowledge of the test’s performance characteristics. Diagnostic tests are evaluated for their ability to correctly classify animals as infected (sensitivity), or as uninfected (specificity) [1], characteristics normally defined by comparing test results with a reference test (sometimes called a gold standard [2,3]). Whilst these parameters may be defined by large-scale studies in domestic species, sample sizes in wildlife are often limited due to difficulties accessing or capturing animals and, in some cases, by ethical considerations [4]. Here, we evaluate the performance of the available diagnostic tests for tuberculosis (TB) in wild meerkats (*Suricata suricatta*) and use combinations of tests and life history information to increase our understanding of the results.

A diagnostic test works by determining the presence or quantity of a relevant biological marker, such as antibodies, in order to assign a positive or negative result. In a chronic disease such as TB, the abundance of these markers may vary over time. Therefore, as TB progresses from initial infection to clinical disease, a particular diagnostic test’s performance may change with the abundance of the targeted biological entity [5,6,7,8]. A test based on measuring presence of bacilli, for example, might work best at a later disease stage when bacterial replication has taken place, rather than at the point of initial infection. This means that a test may perform well at only certain stages of infection, and if relied upon blindly, may be misinterpreted. A combination of tests targeting different biological parameters may better reflect true infection status compared to a single test, especially when applied across a population with individuals at different stages of a chronic disease.

In meerkats, TB is caused by infection with *Mycobacterium suricattae*, a member of the *Mycobacterium tuberculosis* complex [9]. Persistently swollen lymph nodes (particularly the submandibular) are typical signs of advanced disease in this species, leading to a presumptive diagnosis of TB [10]. Transmission has been linked with suppurating skin wounds and oral excretion [11]. Given the highly social behaviour of these animals, these routes mean that infection may easily be transmitted throughout a social group [12]. Diagnostic tests may therefore be justifiably targeted at either the group- or the individual-animal level. Where individual test sensitivity is low, testing multiple individuals gives a more reliable estimation as to whether there is truly infection within a group.

The Kalahari meerkat project (KMP) is a long-term ecological study in the Northern Cape of South Africa [13]. Tuberculosis was first identified at the study site in 1999; since then, clinical signs have been observed in meerkats in the majority of years, affecting 6% of individuals born in the study population [12]. The KMP therefore offers a unique opportunity to study TB and its transmission in a natural setting. Each meerkat is individually identifiable [14], and all animals are habituated to close observation which enables repeated sampling over time. It is the Project’s policy to euthanase meerkats with advanced stages of TB at the point at which lymph node swellings burst, in an effort to control spread of disease. Previous records indicate that animals that have reached this stage and are not euthanased, seldom live for more than a month, and never recover. Given the potential applications of TB research that could be carried out at the site, it is important to have a thorough understanding of how to interpret available diagnostic tests.

Previous TB diagnostic assays used in meerkats have focused on antibody responses and mycobacterial culture [15]. Cultures of tracheal washes, multiantigen print immunoassay (MAPIA), and a commercially available lateral flow immunoassay rapid test (RT) were shown to be of diagnostic value when used in combination, but had limitations in terms of either sensitivity (culture and RT) or specificity (MAPIA) when interpreted independently [15]. More recently, a test of cell-mediated immunity (CMI), based upon interferon-gamma (IFNγ) inducible-protein 10 (IP-10), has been developed for use in meerkats [16], and a new version of the lateral flow device (the dual-path platform, DPP; Chembio Diagnostics Systems, Inc., Medford, New York, NY, USA) has been developed [17]. The sensitivity and specificity characteristics of the IP-10 release assays (IPRA) and the DPP serologic assay are unknown. Due to the known pattern of disease response in other species (e.g., cattle) [18], tests of CMI are typically able to detect infection at an earlier stage than culture or antibody tests.

This study investigated the performance of the IPRA, the DPP, and mycobacterial culture to determine *M. suricattae* infection status both with and without a reference test. We asked how these test results relate to true infection status by investigating associations between tests, individual animal survival, and the likelihood of clinical disease being present. Finally, we determined whether an individual’s age influences its diagnostic test result, and whether this has implications for test reliability. These approaches made use of the same test dataset, but in different ways, giving multiple insights into how the tests may be used and the limitations of the available methods.

## 2. Materials and Methods

### 2.1. Data Collection

Nine meerkat social groups at the KMP were selected at random for longitudinal sampling (Groups B, F, J, N, P, Q, R, U, and Z). From September 2014 to September 2016, 268 meerkats were sampled in five sampling periods (Blocks 1–5), each block lasting for three months. Within each block, all animals within the sample groups were scheduled to be sampled once. In block 1, samples from a further 44 individuals were available for testing from a parallel study [16] involving an additional 6 groups, and samples previously collected (*n* = 10) from a captive population described by Clarke et al. [16], were utilised. Ethical approval for the study was granted by the University of Pretoria’s ethics committee in May 2014 (project reference ECO20-14).

Throughout this two-year study, each group was visited a minimum of three occasions per week. At each visit, all individuals within a group were visually examined for signs of illness or injury. Individuals were identified by both a subcutaneous microchip, and a unique combination of hair dye markings regularly applied to the fur. Presence of each animal was recorded, and an individual was recorded as dead if it had not been seen for a period of three months. Animals were also visually checked for swollen lymph nodes as an indicator of likely disease. Observations contributing to the study therefore continued until 31 December 2016 to allow for this lag. In addition to these life history data, field sampling was carried out in order to collect blood and tracheal wash samples. Captures, anaesthesia, and sampling were performed as described by Drewe [15]. Age was calculated at the point of sampling, based on recorded birth dates for each individual.

Euthanasia of individuals with draining lymph nodes was performed under anaesthesia. Blood samples and tracheal wash samples were collected, after which 2 mL of sodium *pentobarbital* (Euthanase, Kyron, Johannesburg, *South Africa*) was injected into the jugular vein. Post-mortem examinations were performed on all euthanased animals in the field. Samples of liver, lung, spleen, and sub-mandibular lymph nodes were frozen at −80 °C prior to being transported to the laboratory. Further samples were similarly submitted from additional organs and lymph nodes with gross abnormalities on post-mortem examination. All samples were transported on ice for full laboratory processing.

### 2.2. Laboratory Processing

Both tracheal washes and post-mortem tissue samples were processed as previously described (15) and cultured in Mycobacterial Growth Indicator Tubes (MGIT, Becton Dickinson, Franklin Lakes, NJ, USA) for 12 weeks in a biosafety level 3 laboratory (Stellenbosch University). Ziehl-Neelsen stain positive cultures were genotyped by PCR targeting selected regions of difference [9,19].

Bovigam^®^ PC-HP peptide (Prionics, Schlieren-Zurich, Switzerland) was used as a source of *M. bovis* antigens in the cytokine release assay. IP-10 assays were performed as described by Clarke et al. [16]. An adjusted test result was obtained by subtracting the Optical Density (OD) reading for the saline stimulated sample, OD^nil^, from that for the PC-HP stimulated sample, OD^PCHP^. A test cut-off value for OD^PCHP-nil^ of 0.038 was used based on a prior study, and OD^PWM^ was used as a positive control [16]. Serological analysis was performed using a rapid lateral flow test, DPP, using 5 µL of stored plasma, according to manufacturer’s instructions. To quantitate test results as relative light units (RLU), an optical reader (Opticon DPP test reader, Chembio Diagnostic Systems, Inc., Medford, New York, NY, USA) was used to measure the reflectance of a control line and two test lines containing the antigens MPB83 (line 1) and ESAT-6/CFP-10 recombinant protein (line 2). A diagnostic cut-off value of 5.0 RLU was used. Reactivity to either test line in the DPP was considered as a positive antibody result.

### 2.3. Data Analyses

Each ante-mortem test result was expressed as either positive or negative as described above, and a parallel interpretation was added, which was defined as positive if one or more of the three individual tests (serology, IPRA, or culture) returned a positive result. An “ever positive” category was also created in which an animal was deemed negative (uninfected) until its first positive parallel test result, after which it remained categorised as positive at all subsequent time points. The motivation for each analysis, and the samples included in each, are outlined in Table 1.

### 2.4. Sensitivity Estimates in the Presence, and Absence, of a Reference Test

Sensitivity of each test (ability to correctly identify infection) was estimated both in the presence and absence of a reference test (Table 1, Aims 1 and 2) and a kappa statistic was calculated to estimate test agreement (Table 1, Aim 3). For latent class analysis, three populations with different TB prevalences were defined: (1) 0% prevalence—a negative control population (*n* = 10) consisting of the previously sampled captive animals [16] with no TB disease history; (2) unknown prevalence—samples from the wild population, with unknown prevalence (*n* = 171); and (3) 100% prevalence—a positive control population consisting of wild animals, confirmed to be diseased following post-mortem cultures (*n* = 11). In these populations, the cross-detection of meerkats with the three different diagnostic tests (serology, IPRA, and culture) was modelled following a latent class modelling approach. The analysis was implemented in a Bayesian framework using the WinBUGS software (MRC, Cambridge, UK, [28]) embedded within the R package (R Core Team 2014) using the R2WinBugs library [29]. Uniform distributions (0,1) were assumed for all sensitivity and specificity priors. A prior for the wild population prevalence was given by the beta (1.5, 2.3) distribution, to fit with a prior estimation of approximately 30% prevalence. Diagnostic tests were assumed to be independent due to their differing biological mechanisms of action. Two simulation chains of 1,000,000 iterations were run, with the first 100,000 iterations discarded to allow for burn-in of the chain. The chains were then thinned, taking every hundredth sample to reduce autocorrelation among the samples. Convergence was assessed by checking the trace plots for all monitored parameters [30].

### 2.5. Survival Analyses

Time-dependent Cox regression [25,31] models were constructed in R using the Survival package [26] to evaluate time to death, and time to the appearance of clinical signs. Individuals entered analysis either at the start of the study period, 1 September 2014, or at their date of birth if this was later. Each individual’s time in the study was divided into time periods from the time of one sampling event until the next. The event “death” was created to describe either the point of death where this was observed, or the date that the animal was last seen if it had not been recorded for a minimum of three months. Within each time period, the animal’s age (categorised as under months, 6–12 months, or >12 months), sex, dominance status, social group, and test statuses were recorded. A univariable analysis was carried out to examine the effects of age, dominance status, sex, and test status on the time to two events: appearance of a persistently enlarged lymph node, and loss of the animal from the study population. For time to loss, analysis was carried out by both including and excluding animals that were euthanased as this intervention had an impact upon the end time point, and the time of the final sampling point. When analysing the time until the appearance of a swollen lymph node, animals that had been observed with a swelling prior to their first sampling episode were excluded. 

### 2.6. Relationships with Time, Age, and Predicted Risk

All individuals were assigned to one of four risk categories based upon age and group history of disease, using the findings outlined by Patterson et al. [12]. Descriptive statistics were calculated to describe how test results related to risk category, how they changed for each individual that was sampled repeatedly over time, and the relationship of test status with age.

## 3. Results

### 3.1. Sensitivity Estimates in the Presence, and Absence, of a Reference Test

Over the course of the study, *M. suricattae* was cultured from post-mortem samples of 23 meerkats, in six different social groups (Table 2). Diagnostic test results were available for all three tests for 22 of these animals, and of these, parallel test interpretation was the most sensitive measure, correctly identifying 59% (95% confidence interval [95CI]: 37–79%) of culture-positive animals. Sensitivity estimates were greater when tests were carried out closer to the time of death. At all time points, previous test history increased sensitivity, with 82% (95CI: 59–95%) of culture-positive individuals having tested positive at least once on a minimum of one diagnostic test. Limited sample size led to wide confidence intervals when evaluating individuals further away from the euthanasia time point (Table 2).

On 174 occasions, contemporaneous results were obtained for all three of the diagnostic tests. There were no sampling events where all three tests agreed on a positive diagnosis, whilst all three were negative on 103 occasions (59.2%). As only a single tracheal wash cultured positive for *M. suricattae* (an IP-10 positive animal with a negative DPP result), culture was excluded from the kappa statistic calculations. The kappa statistic for agreement between the serology and the cell-mediated immunity test results was 0.009 (95CI: 0–0.19).

When analysed in the absence of a reference test (Table 1, Aim 2), the IPRA had the highest sensitivity of the three tests (58.6%; 95% Credible Interval [95Cred]: 35.1–80.6%), and the culture had the greatest specificity (99.2%, 95Cred: 95.7–99.9%) (Table 3). Using a parallel interpretation of three tests, sensitivity increased to 65.7% (95Cred: 42.7–84.6%), at the expense of a fall in specificity to 81.9% (95Cred: 65.2–96.7%) (Table 3).

### 3.2. Survival Analyses

A total of 126 animals were included in the survival analysis (Table 1, Aims 4 and 5). Fourteen animals from this cohort were euthanased, a further 104 were lost to follow up and assumed dead, and the remaining animals lived beyond the study period. Clinical signs of TB were observed in 22 (17.5%) of these individuals. There was no evidence of an effect of age, sex, nor dominance status on the survival time (Table 4). Strong evidence was found for the effects of serology and culture test statuses (*p* = 0.04 and *p* = 0.01, respectively), and for the result of parallel test interpretation (*p* = 0.01), both when including and excluding euthanased animals. Only very weak evidence was found for an effect of the CMI-test status (*p* = 0.11). However, the multivariable model showed evidence of confounding, as the significance of the diagnostic test parameters disappeared with the addition of sampling groups, suggesting that the two were associated (Table 4). The Cox proportional hazards assumption was met for all eight models. The analysis of time until an observed persistently enlarged lymph node revealed a hazard ratio of 3.91 (95%CI: 0.89–17.23, *p* = 0.07) for individuals that were positive on the serological test (Table 1).

### 3.3. Relationships with Time, Age, and Predicted Risk

For 44 individuals, IPRAs were performed on more than one occasion (range 2–5, median, three tests per individual). Of 29 individuals that initially tested negative, 15 (52%, 95CI: 33–70%) became positive at a subsequent test. Ten animals (66.7%, 95CI: 39–87%) initially testing positive, tested negative at a later test. In six of these events (60%), the test result was influenced by a high OD^nil^ result in the upper quartile of all OD^nil^ values.

Serology was performed on multiple occasions (range 2–6, median 3) for 52 individuals. Four individuals (8%, 95CI: 3–19%) converted from negative to positive over a pair of sampling points. Only a single positive animal was presented for retesting at a later time point—this individual subsequently tested negative.

For 44 individuals, there were multiple sampling points (range 2–5, median 3), at which all three diagnostic tests were performed (Figure 1). Using a parallel test interpretation, nine out of 15 positive individuals (69%, 95CI: 33–83%) converted to a negative result (on all three tests) between consecutive tests (Table 1, Aim 6). Post-mortem examinations of those 15 individuals were not possible.

No association between age and test result (Table 1, Aim 7) was found in the data from 704 serological tests (Appendix A). Confidence intervals for prevalence in the 0–3 month age category did not overlap with the 9–12, the 12–15, nor the 18–21 month categories, but no other such differences were observed. An increase in the mean OD^PCHP-nil^ was observed between the 2014 and 2015 seasons and 2016 for one group (group N in Appendix A), but at all other times, there were no differences between results over time.

The predicted-risk analysis (Table 1, Aim 8) incorporated 159 sampling events, at which 153 serology results were obtained, 112 IPRA results, and 159 culture samples. No differences were observed in the proportions testing positive for either culture or serology between any of the four risk categories (Figure 2). Parallel interpretation and IPRA both differentiated between the categories with 100% (95CI: 62.9–100%) prevalence in the highest risk category (both risk factors) and 17.0% (95CI: 8.5–30.3%) and 15.1% (95CI: 7.2–28.1%), respectively in the lowest risk category (neither risk factor).

## 4. Discussion

This work used a range of techniques to investigate the performance of three diagnostic tests (IPRA, DPP, and mycobacterial culture) in order to better understand how best to utilise them in studies of tuberculosis in meerkat populations. While addressing this, we highlight characteristics of diagnostic tests that should remind practitioners of the care needed when interpreting such data. The results demonstrate that through the use of multiple methods in combination, confidence can be obtained in diagnostic tests, despite an absence of controlled environments.

### 4.1. Variation in Results

There were no occasions at which all three diagnostics tests were positive, and overall test agreement was low (Kappa statistic for agreement between serology and IPRA was 0.009, 95CI: 0–0.19), likely due to the tests detecting different stages of disease. The tests identify discrete stages of the disease process [18], thus a positive cell-mediated response does not make a contemporaneous serological or culture result any more likely. In addition to discrepancy between contemporaneous results, there was a trend toward increased sensitivity the closer an animal was to the end-stage of disease (Table 2). It is important that practitioners are aware of these differing test performances and consider these when interpreting results, not just tuberculosis in meerkats, but chronic diseases across a range of species. Failure to do so will lead to underestimations of disease prevalence.

When interpreting results, consideration should also be given to whether the research question is targeted at diagnosis of an individual or a group. Comparison of IPRA results for four social groups (Appendix A) demonstrated a shift in test results in one group (N) shortly before disease was observed in the group in late 2016. Wide variation in results in two social groups is also of interest. The pattern in group Z in both 2014 and 2015 is similar to that seen in group F shortly before the latter group died out due to clinical TB. No disease was observed in group Z, but high numbers were lost coinciding with a drought in late 2015, and these results may possibly reflect a role of sub-clinical infection interacting with harsh environmental conditions. A substantial proportion of the individuals in one group (group Z) were older animals, and one speculated reason for the higher IPRA results in that group in 2014–15 may have been an association between age and antigen-specific cell-mediated immunity. However, across all samples, no association was found between an individual’s age, and their IPRA result. Combining the IPRA results of individuals appears to hold promise for interpreting group-level disease status.

### 4.2. Test Performance

Of the individual diagnostic tests used, the IPRA was the most sensitive (45%, 95CI: 25–67%, compared with the reference test and 58.6%, 95Cred: 35.1–80.6% by Bayesian estimation) and least specific (Bayesian estimation: 85.0%, 95Cred: 68.4–99.4%) of the three diagnostic tests used. Inducible-protein 10 studies in cattle and human have shown very high specificities of 100% and 97% respectively, with sensitivities of 96% and 81%, [32,33] much higher than seen in this study. Sensitivity of the IPRA in meerkats was reduced by spontaneous release of IP-10 in the unstimulated samples, a phenomenon previously reported in cattle [32]. Again, considered alone, serological sensitivity estimates in this study were low. A range of sensitivity estimates for TB serology in wildlife are present in the literature [34,35], and the estimates here (12%, Table 3) fall well below published data. Using the same serological test, sensitivities and specificities have been found to be 29.5% and 88% in European badgers (*Meles meles*), 65.1% and 97.8% in white-tailed deer (*Odocoileus virginianus*), and 57.5% and 96.7% in alpaca (*Vicugna pacos*), respectively [36,37,38].

We asked how well the three available tests were able to correctly identify infected and uninfected meerkats (sensitivity and specificity). Two methods were used to estimate sensitivity of the parallel test; testing against a reference test (59%; 95CI: 37–79%), and a Bayesian approach in the absence of knowledge about true infection status (65.7%; 95Cred: 42.7–84.6%). Only a small number of confirmed TB cases were available for the reference test, thus there was benefit in having a second, similar, estimate of sensitivity from a different method. It was estimated that the specificity of this interpretation was 81.9% (95Cred: 65.2–96.7%) by using a Bayesian approach. Specificity could potentially have been reduced by cross-reactivity with other *Mycobacterium* spp., which may be present in this environment. Parallel interpretation improved sensitivity; serial interpretation (requiring all results to be positive in order to classify the animal as infected) was not examined, having previously been shown to be extremely limited in TB diagnostics [39]. While interpreting results in a series will always bring about some level of reduction in sensitivity, it is particularly likely to do so when the tests are detecting different stages of disease and have poor test agreement.

### 4.3. Prognostic Indicators

Gold standard validation of a diagnostic test compares results to an accepted standard. In the medical literature, the importance of measuring tests’ performances against clinical characteristics has recently been highlighted [40], and this study used prognostic indicators as a proxy for confirmed disease presence. Two methods were used to assess how results obtained related to outcomes.

Firstly, the results, from those animals considered most likely to be infected, were checked to see if more positive test results were obtained than for animals at greater risk. Patterson et al. [12] showed that increasing age, and a previous history of disease within a social group, increased the risk of visual signs of clinical TB. It was therefore expected that those at risk animals would have the greatest levels of infection. The proportion of animals testing positive on IPRA was greatest when two risk factors were present (increased age and group history), and least when no factors were present (Figure 2). The parallel interpretation was heavily influenced by the IPRA results, whereas serology and culture failed to differentiate between the risk categories. This suggests that the IPRA is the more useful of the individual tests for diagnosis at a group level.

Secondly, in terms of providing meaningful results, some evidence has been provided for an association between survival and both serology and culture results (*p* = 0.038 and *p* = 0.013, respectively, Table 4). This relationship disappears in multivariable analysis, almost certainly as a result of the confounding effect of social group; it is likely that infection statuses are shared between members of these socially intimate animals. Relative to the IPRA, stronger evidence was found for serology and culture results being associated with survival, with hazard ratios of 2.4 and 14.8, respectively. This suggests that positive status for these tests might be associated with end-stage disease, as positive results occur shortly before the end of an animal’s life. Serology was the only variable where any relationship with time to the appearance of persistently swollen lymph nodes was observed, and the evidence was weak (*p* = 0.072, Table 1). Evidence is likely to be weak due to the small number of animals testing positive on this test. Observer bias is unlikely to have influenced the recording of these swollen lymph nodes, as the observers were not aware of test results.

## 5. Conclusions

Given the characteristics of the three tests, there are multiple ways in which their results could be interpreted, depending upon specific research or management questions. Culture and serology were estimated to have the greatest specificity, be the best predictors of all-cause mortality, and there may also be a weak association with the appearance of clinical signs. These findings were similar to studies in cattle showing that antibodies and bacterial load tend to increase only late in TB pathogenesis [6,8]. Rua-Domenech et al. [18] stated that serological tests were not an alternative to CMI-based tests in cattle, but rather could be used as a confirmatory test in parallel with the skin test. Similarly, in the current study, serology and culture appear to be specific, confirmatory tests of disease. However, the sensitivity of these tests is very poor and visual reporting of disease signs identify many more cases than these tests combined. As a surveillance tool, diagnostic assays which are less sensitive than visual signs provide little advantage. In contrast, the IPRA appears to have greater sensitivity and would be a better surveillance tool for infection. Even in cattle, where far greater resources have been put into diagnostic test development and ante-mortem surveillance, post-mortem testing at slaughterhouses can still make a significant contribution to the identification of cases missed ante-mortem [41,42]. There may be limitations using the IPRA in late-stage disease; therefore, it should be interpreted in parallel with the other tests to optimise overall sensitivity, an interpretation shown to be effective in studies of other species [43]. Parallel interpretation involves a loss of specificity but given the high specificities of the serology and culture, this reduction appears to be small. While greater sensitivities and specificities have been found for well-studied species such as badgers, humans, cattle, bison and deer, the estimates for the combined tests of 65.7% and 81.9% are comparable to other published test performance statistics [34]. For further studies of tuberculosis in meerkats focussing on individuals, a parallel interpretation of these three diagnostic methods is recommended.

Specific recommendations have been made within this study for how best to use combinations of currently available diagnostic tests when studying TB in meerkats. Weaknesses in individual test performances, and the complexities of this chronic disease contribute to potential pitfalls in interpretation, which have been highlighted. An important message from this study is that despite imperfect conditions, valuable information can be obtained about test performance when utilising multiple different methods of evaluation and incorporating outcomes into those analyses. Such approaches are recommended when working with wildlife datasets, which, due to real-world conditions, are often collected with imperfect sampling methods, using unvalidated tests, and yet are uniquely valuable.

## Figures and Tables

**Figure 1 animals-11-03453-f001:**
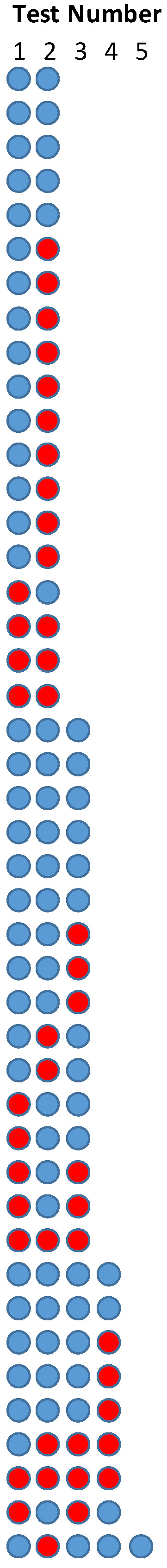
Parallel testing (using culture, serology, and a test of cell-mediated immunity) of 44 individuals tested on multiple occasions. Each row represents an individual animal, and each circle, a separate sampling point. Positive tests are shown in red, and negative in blue. The sampling points are shown in the order in which they were collected, with the earliest on the left, and the most recent sample shown on the right. Although the time between sampling points varied, the minimum interval between two samples was three months. Nine individuals changed from positive to negative test status within the two-year study period; three of these subsequently reverted to a positive test status.

**Figure 2 animals-11-03453-f002:**
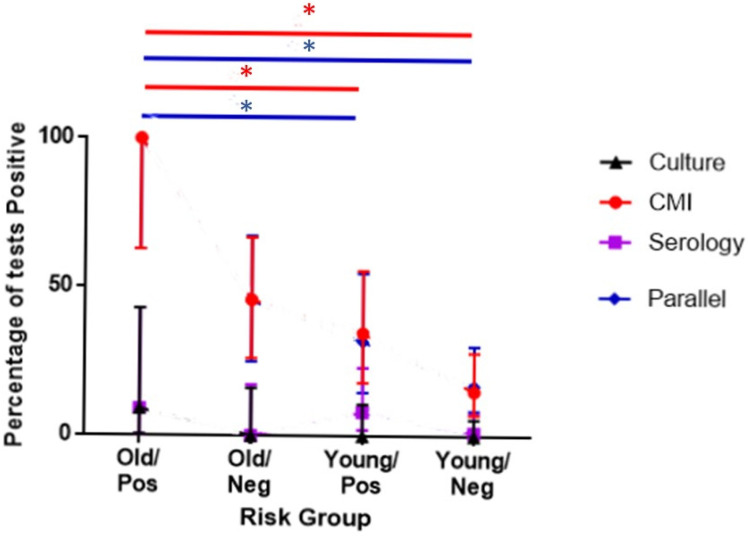
Association between predicted risk of infection and diagnostic test results. The percentage of animals testing positive to each test is shown in each of four risk categories, with 95% confidence intervals. Risk categories are defined by age (animals aged over 2 years were categorised as Old, and those below 2 years as Young) and social group history of disease (groups having had clinical cases being “Pos”, and those without being “Neg”). Culture results are based upon a tracheal wash culture, cell-mediated immunity (CMI) results are those from the IPRA (IP-10 release assays), and serology results are based upon the DPP (dual-path platform) assay. A test was considered positive on parallel test interpretation if it tested positive to one or more of the individual tests. The results of tests of cell-mediated immunity and the parallel test interpretation reflect an individual’s risk of being infected, whereas no difference is seen between risk groups when testing with serology or culture. As indicated by the *, for both the CMI and the parallel tests, there is a significant difference (*p* < 0.05) between the highest risk group (Old/Pos) and both the Young/Pos and the Young/Neg groups.

**Table 1 animals-11-03453-t001:** Inclusion criteria and justification for each analysis conducted as part of this study.

Aim of the Analysis	Approach Used	Inclusion Criteria	Time Period	References
1. To determine the ability of each testing regime to correctly identify infected animals (test sensitivity)	Comparison with a reference test	All animals from which *M. suricattae* was confirmed on culture of post-mortem material, at any point during the study. All samples collected prior to death were included.	September 2014-September 2016 inclusive (Full)	[3]
2. To determine each testing regime’s ability to correctly identify animals as infected (sensitivity) or uninfected (specificity)	Latent Class Analysis	The first sample collected for each individual was used. The time-period was limited to minimise the impact of changes over time on prevalence.	September 2014–March 2015 inclusive	[20,21,22,23]
3. To quantify the agreement between the results from different tests	Calculation of the kappa statistic	All sampling events where there was more than one test were used.	Full	[24]
4. To investigate whether test results were predictive of mortality	Survival analysis of time to death	All samples collected over the study period from wild individuals were used.	Full	[25]
5. To investigate whether test results were predictive of clinical signs of disease	Survival analysis of time to showing clinical signs	All samples collected over the study period from wild individuals were used.	Full	[25]
6. To investigate whether individuals’ test results fluctuate over time	Analysis of consistency over time	All individuals sampled on more than one occasion were included.	Full	-
7. To investigate whether test result is influenced by age	Analysis for association with age	All samples from individuals with a known date of birth were included.	Full	-
8. To Investigate whether test results were detecting infection more often in the most susceptible animals	Analysis for association with risk	Only the first sample from individuals with a known date of birth were included. The time-period was limited to minimise the impact of changes over time.	September 2014–March 2015 inclusive	[12]

Variables found to have a significance of *p* < 0.2 in the univariable analysis were incorporated into a multivariable analysis in a forward-stepwise process, with each addition to the model compared to the simpler version using analysis of deviance for a Cox model [26]. A gamma-frailty term was included for the social group in which the individual was resident [27].

**Table 2 animals-11-03453-t002:** Sensitivity estimates for three diagnostic tests, when used alone and in combination, compared to a reference test. Ante-mortem test results are presented for 23 individuals that were later confirmed as definitively infected by post-mortem culture. Animals were tested using tracheal wash culture, DPP assay, and IFNγ inducible-protein 10 release assays (IPRA). All 23 animals were tested at the point of euthanasia, and results from previous sampling points are provided at time points relative to euthanasia. An animal was considered positive on parallel interpretation if it had a positive result on at least one of the three individual tests. A positive status to the “ever positive” interpretation was given if any single individual test was positive either at that time point or ever in the animal’s test history.

		Individual Test Interpretations			Multiple Test Interpretations	
Time before Euthanasia (days)	*n*	Serology (DPP) (%, 95CI *)	Cell-Mediated Immunity (IPRA) (%, 95CI *)	Tracheal Wash Culture (%, 95CI *)	Parallel (%, 95CI *)	Ever Positive (%, 95CI *)
0	23	3/22 (13, 3–35)	10/22 (45, 25–67)	3/23 (13, 3–35)	13/22 (59, 37–79)	18/22 (82, 59–95)
1–90	10	0/10 (0, 0–35)	4/10 (40, 14–73)	0/10 (0, 0–35)	4/9 (44, 15–77)	6/9 (67, 31–91)
91–180	7	0/7 (0, 0–44)	4/7 (57, 20–88)	0/7 (0, 0–44)	4/7 (57, 20–88)	5/7 (71, 30–95)
181–365	11	0/11 (0, 0–32)	1/8 (13, 1–53)	0/11 (0, 0–32)	1/8 (13, 1–53)	2/8 (25, 5–64)
>365	11	0/11 (0, 0–32)	1/8 (13, 1–53)	0/11 (0, 0–32)	1/8 (13, 1–53)	1/8 (13, 1–53)

* 95CI, 95% confidence interval. DPP: dual-path platform; IPRA: IP-10 release assays.

**Table 3 animals-11-03453-t003:** Bayesian sensitivity and specificity estimates for three diagnostic tests, when used alone and in combination, in the absence of a reference test. The model for these estimates was based upon tests taken from three populations; a captive population with assumed zero prevalence (*n* = 10), the wild population of unknown prevalence (*n* = 171), and a group of post-mortem confirmed cases with 100% prevalence (*n* = 11). The dual-path platform VetTB (tuberculosis) assay was used for serology, an IP-10 (inducible-protein 10) release assay for the test of cell-mediated immunity, and a tracheal wash sample was provided for culture. The parallel interpretation was considered positive if one or more of the individual tests was positive. The model estimated a prevalence of 32.2% (95% confidence interval: 4.8–69.2%) in the wild population. Convergence of the Bayesian model was considered good.

	Serology (DPP)	Cell-Mediated Immunity (IPRA)	Tracheal Wash Culture	Parallel Interpretation
Sensitivity (%, Credible interval)	12.0 (4.0–33.6)	58.6 (35.1–80.6)	2.2 (0.1–10.9)	65.7 (42.7–84.7)
Specificity (%, Credible interval)	97.9 (92.8–99.9)	85.0 (68.4–99.4)	99.2 (95.7–99.9)	81.9 (65.2–96.7)

DPP: dual-path platform; IPRA: IP-10 release assays.

**Table 4 animals-11-03453-t004:** Survival analysis of time to death. Results of univariable and multivariable analysis using time-dependent Cox regression. Hazards were calculated for the likelihood of a meerkat being lost from the study population. Model (a) includes all individuals (of whom 118 experienced the event, death), and on the following page, model (b), excludes euthanased individuals.

		**Model a—All Animals (*n* = 126)**			
**Univariable**		**Hazard Ratio**	**95% Confidence Interval**	***p*-Value**	
Dominance	No			0.966	
	Yes	0.99	0.64–1.53		
Sex	Female			0.813	
	Male	1.05	0.711.54		
Age	<6 months			0.678	
	6–12 months	1.03	0.55–1.92		
	>12 months	0.85	0.49–1.45		
Serology	Neg ^a^			0.039	
	Pos ^b^	2.44	1.04–5.70		
Cell-Mediated immunity	Neg ^a^			0.107	
	Pos ^b^	1.45	0.92–2.28		
Culture	Neg ^a^			0.013	
	Pos ^b^	14.84	1.76–125.5		
Parallel	Neg ^a^			0.010	
	Pos ^b^	1.75	1.14–2.67		
Multivariable		Hazard Ratio	95% Confidence Interval	*p*-value	Likelihood ratio for model, *p* value
Serology + Frailty Sampling group	Neg ^a^				<0.0001
	Pos ^b^	1.29	0.51–3.31	0.59	
Variance of frailty term		1.126			
Cell-Mediated immunity + Frailty Social group	Neg ^a^				<0.0001
	Pos ^b^	1.10	0.65–1.87	0.72	
Variance of frailty term		1.939			
Culture + Frailty Social group	Neg ^a^				<0.0001
	Pos ^b^	1.92	0.22–16.60	0.55	
Variance of frailty term		1.075			
Parallel + Frailty Social group	Neg ^a^				<0.0001
	Pos ^b^	1.30	0.80–2.11	0.29	
Variance of frailty term		1.028			
		Model b–Excluding euthanised animals (*n* = 112)			
**Univariable**		Hazard Ratio	95% Confidence Interval	*p*-value	
Dominance	No			0.780	
	Yes	1.06	0.69–1.65		
Sex	F			0.820	
	M	1.05	0.70–1.56		
Age	<6 months			0.897	
	6–12 months	0.93	0.48–1.79		
	>12 months	0.88	0.51–1.53		
Serology	Neg ^a^			0.116	
	Pos ^b^	2.09	0.83–5.27		
CMI	Neg ^a^			0.09	
	Pos ^b^	1.48	0.93–2.34		
Culture	Neg ^a^			0.008	
	Pos ^b^	18.66	2.13–163.60		
Parallel	Neg ^a^			0.011	
	Pos ^b^	1.75	1.14–2.68		
**Multivariable**		Hazard Ratio	95% Confidence Interval	*p*-value	Likelihood ratio for model, *p* value
Serology + Frailty Sampling group	Neg ^a^				<0.0001
	Pos ^b^	1.04	0.38–2.86	0.94	
Variance of frailty term		1.126			
CMI + Frailty Social group	Neg ^a^				<0.0001
	Pos ^b^	1.17	0.68–1.99	0.57	
Variance of frailty term		1.809			
Culture + Frailty Social group	Neg ^a^				<0.0001
	Pos ^b^	2.37	0.26–21.39	0.44	
Variance of frailty term		1.013			
Parallel + Frailty Social group	Neg ^a^				<0.0001
	Pos ^b^	1.33	0.81–2.17	0.26	
Variance of frailty term		0.972			

^a^ Neg, Negative. ^b^ Pos, Positive.

## Data Availability

Data presented in this study is publicly available at https://figshare.com/s/45bd5369a97b4c2489bf (accessed on 25 October 2021).

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
