# Peer review of "Combining Analytical Approaches and Multiple Sources of Information to Improve Interpretation of Diagnostic Test Results for Tuberculosis in Wild Meerkats"

_animals, 2021, doi:10.3390/ani11123453_

Round 1

Reviewer 1 Report

The manuscript is overall good and addresses an interesting topic for researchers studying infectious diseases of wild animals. The researchers presented the work in a clear, logical and understandable way to the reader. The methodological and statistical approach is solid and the simple size appears adequate.

I believe that the work is solid and useful to the scientific community and should be accepted by the journal for its publication.

I recommend reviewing the entire bibliography as the reference to line 218 (Patterson et al. [31]) does not correspond with the reference cited (Hougaard P. Frailty models for survival data. Lifetime Data Analysis. 1995;1(3):255-73. doi: 10.1007/BF00985760).

Author Response

The manuscript is overall good and addresses an interesting topic for researchers studying infectious diseases of wild animals. The researchers presented the work in a clear, logical and understandable way to the reader. The methodological and statistical approach is solid and the simple size appears adequate.

I believe that the work is solid and useful to the scientific community and should be accepted by the journal for its publication.

I recommend reviewing the entire bibliography as the reference to line 218 (Patterson et al. [31]) does not correspond with the reference cited (Hougaard P. Frailty models for survival data. Lifetime Data Analysis. 1995;1(3):255-73. doi: 10.1007/BF00985760).

Response: Thank you for reviewing this work and for those positive comments. With regards to the referencing, this observation is helpful, thank you. We have amended that particular reference (now line 223) and checked through the remainder of the manuscript, thank you.

Reviewer 2 Report

This study aimed to combine the performance of three available ante-mortem diagnostic tests (IRPA, DPP, mycobacterial culture) for determine tuberculosis infectious status in wild meerkats both individually and in combination, also comparing tests results with post-mortem findings, survival times and clinical characteristics to quantify the prognostic capability of the tests. Overall, this study is very interesting and provides useful information concerning the diagnostic limitations of the currently available ante-mortem tests in diagnosing infectious diseases  in wild animal populations. However, some clarifications are needed especially in the materials and methods section to make the interpretation of the results obtained clearer, and all main concerns are listed below. 

Line 38. In the Abstract, Authors write: "over a two-year period, we sampled 166 free-living meerkats"; However, this information is in contrast with that reported in the Methods section (Line 121), where Authors write about 268 animals sampled from September 2014 to September 2016. 

Lines 41-42. Authors write: "Post-mortem cultures confirmed M. suricattae infection in 26 animals, 59% of which were..." and this information does not agree with those reported in the results (Lines 225-226). 

Lines 120-126. Information concerning meerkats social groups, sampling periods and blocks should be explained more clearly. In particular, in the discussion section, Authors mention 4 social groups (F, N, Q and Z) which have not been explained in the methods section. 

Lines 434-446. The possibility of false-negative results in the case of TB represents an issue of concern for domestic and wild animal species, including cattle. Through the section falls within the topic of the study, Authors should enhance this section adding more information concerning the importance of post-mortem inspection at the slaughterhouse as essential procedure to obtain reliable information on bovine tuberculosis (bTB) prevalence in cattle in epidemiological surveillance programs. Noteworthy, post-mortem inspection allow the confirmation of bTB in herd test reactors but often provides additional data concerning infected animals which have note reacted in field tests. 

On this regard, Authors could write: "Despite the wide availability of ante-mortem tests for identification of Mycobacterium bovis infection at herd level, diagnosis of bTB is difficult often because of the scarse of diagnostic tests that fulfills all the essential criteria necessary for identification of infected animals and noteworthy, almost the 20 % of the new bTB cases are firstly diagnosed during post-mortem inspection at the slauaghterhouse in cattle intended for human consumption (Abbate JM et al., 2020; Pascual-Linaza et al., 2017). 

Abbate JM, Arfuso F, Iaria C, Arestia G, Lanteri G. Prevalence of Bovine Tuberculosis in slaughtered cattle in Sicily, Southern Italy. Animals. 2020, 10(9):1473. 

Pascual-Linaza AV, Gordon AW, Stringer LA, Menzies FD. Efficiency of slaughterhouse surveillance for the detection of bovine tuberculosis in cattle in Northern Ireland. Epidemiol. Infect. 2017, 145:995-1005. 

Author Response

This study aimed to combine the performance of three available ante-mortem diagnostic tests (IRPA, DPP, mycobacterial culture) for determine tuberculosis infectious status in wild meerkats both individually and in combination, also comparing tests results with post-mortem findings, survival times and clinical characteristics to quantify the prognostic capability of the tests. Overall, this study is very interesting and provides useful information concerning the diagnostic limitations of the currently available ante-mortem tests in diagnosing infectious diseases  in wild animal populations. However, some clarifications are needed especially in the materials and methods section to make the interpretation of the results obtained clearer, and all main concerns are listed below. 

Thank you for taking the time to read our submission and for these encouraging comments. We have addressed all of your points, as described below, and believe that these clarifications will improve the manuscript.

Line 38. In the Abstract, Authors write: "over a two-year period, we sampled 166 free-living meerkats"; However, this information is in contrast with that reported in the Methods section (Line 121), where Authors write about 268 animals sampled from September 2014 to September 2016. 

Response: Thank you for pointing out that inconsistency. The information in the Methods section is correct, and line 39 has been amended to read “268 free-living meerkats”.

Lines 41-42. Authors write: "Post-mortem cultures confirmed M. suricattae infection in 26 animals, 59% of which were..." and this information does not agree with those reported in the results (Lines 225-226). 

Response: Thank you for pointing out that inconsistency. The information in the Results section is correct, and lines 43-44 has been amended to read “Post-mortem cultures confirmed Mycobacterium suricattae infection in 22 animals which had prior ante-mortem information, 59% (13/22) …”.

Lines 120-126. Information concerning meerkats social groups, sampling periods and blocks should be explained more clearly. In particular, in the discussion section, Authors mention 4 social groups (F, N, Q and Z) which have not been explained in the methods section. 

Response: We agree that some clarification is useful here. We have edited lines 124-125 to clarify two aspects of this methodology. Firstly, we have listed the nine groups names which are later referred to later on in discussion. Secondly, we have clarified the block numbering system which we believe gives the necessary context to mention of “block 1” in line 128.

Lines 434-446. The possibility of false-negative results in the case of TB represents an issue of concern for domestic and wild animal species, including cattle. Through the section falls within the topic of the study, Authors should enhance this section adding more information concerning the importance of post-mortem inspection at the slaughterhouse as essential procedure to obtain reliable information on bovine tuberculosis (bTB) prevalence in cattle in epidemiological surveillance programs. Noteworthy, post-mortem inspection allow the confirmation of bTB in herd test reactors but often provides additional data concerning infected animals which have note reacted in field tests. 

On this regard, Authors could write: "Despite the wide availability of ante-mortem tests for identification of Mycobacterium bovis infection at herd level, diagnosis of bTB is difficult often because of the scarse of diagnostic tests that fulfills all the essential criteria necessary for identification of infected animals and noteworthy, almost the 20 % of the new bTB cases are firstly diagnosed during post-mortem inspection at the slauaghterhouse in cattle intended for human consumption (Abbate JM et al., 2020; Pascual-Linaza et al., 2017). 

Abbate JM, Arfuso F, Iaria C, Arestia G, Lanteri G. Prevalence of Bovine Tuberculosis in slaughtered cattle in Sicily, Southern Italy. Animals. 2020, 10(9):1473. 

Pascual-Linaza AV, Gordon AW, Stringer LA, Menzies FD. Efficiency of slaughterhouse surveillance for the detection of bovine tuberculosis in cattle in Northern Ireland. Epidemiol. Infect. 2017, 145:995-1005. 

Response: Thank you for highlighting this comparison. We have added information to lines 445-447 to point out the importance of post-mortem surveillance and have included the suggested references.

Reviewer 3 Report

Very few comments as attached

Author Response

Thank you for taking the time to review our submission. We found your comments useful and have addressed each point individually below.

Lines 24 and 39: you suggest diagnosing and sampling for ‘tuberculosis’. This might imply the clinical disease. Is it not true that your tests were aiming to identify mycobacterial infection rather than disease? You then use the tests to predict disease outcomes.

Response: We agree that we were testing for the presence and/or response to infection, as opposed to presence of the disease, and thus we have edited lines 24-25 and 40 as recommended.

Line 108: maybe surprised you use cattle, rather than say badgers as an example here.

Response: We agree with the reviewer that there were examples from other species that could have been utilised here, but we felt that the paper by de la Rua-Domenech et al. (2006), which focuses on cattle, describes the variation in immune responses particularly well, and so chose to use this example

Line 110: you use ‘VetTB DPP’ here but not on line 104. Should line 104 also be VetTB DPP to avoid any confusion? Also see comments on lines 159/164 below.

Response: Thank you for pointing out the inconsistency here. We now mention this test for the first time on line 107 and reference the test fully at this point. At this point we describe the abbreviation, DPP, and have amended all further references to the test in the manuscript to simply “DPP”.

Line 136: It seems a bit odd that euthanasia technique (line 140) is described in detail, but anaesthesia technique is all referred to an earlier paper. Perhaps euthansia of meerkats has not previously been described? Some simple description of anaesthesia, even just the drugs used, would seem of interest to most readers. Appreciate some of the methodology is elsewhere though and that repeating that would add to the word count and perhaps be unnecessary.

Response: We have reviewed this section of the text. Both anaesthesia and euthanasia are described in the cited paper [15], and so we have only briefly mentioned both here. Although the details of the techniques may be of interest to readers, we feel that we have provided the appropriate reference to be able to read about these more fully.

Lines 159 and 164: VetTB DPP or DPP VetTB (cf line 110)? I think it’s DPP first, so ‘DPP® VetTB’. Also trademark correctly used here (on the DPP bit) but not in line 104, 106 or 110. This is a bit of a recurring theme through the paper - suggest check all mentions and make it consistent.

Response: As described above, we have amended references to DPP. We have checked a paper (Greenwald et al., 2009) published by an author working at the manufacturer (Chembio) and the trademark symbol is not used in relation to the test in that publication, and so we did not feel it appropriate to do so here.

Greenwald, R., Lyashchenko, O., Esfandiari, J., Miller, M., Mikota, S., Olsen, J.H., Ball, R., Dumonceaux, G., Schmitt, D., Moller, T. and Payeur, J.B., 2009. Highly accurate antibody assays for early and rapid detection of tuberculosis in African and Asian elephants. Clinical and Vaccine Immunology16(5), pp.605-612.

Line 161: Does the optical reader have a name, is it one of the specific Chembio products? I think they have developed new ones on the back of Covid tests, so would be good to say which one you used if was a specific model.

Response: We have added the full name of the optical reader to line 167 in order to clarify this.

Line 206: assume the lymphadenopathy was the only significant clinical sign? Is this mentioned enough in the methodology? There’s not really a mention in that of any physical clinical examination or observation for LN enlargement and/or any other clinical signs. Is it always submandibular LNs only that are considered? How was ‘swollen’ determined?

Response: We have amended lines 78-79 to indicate that although the most commonly affected lymph node is the submandibular, others are occasionally affected: “Persistently swollen lymph nodes (particularly the submandibular)”. Throughout the remainder of the manuscript we have removed specific references to “submandibular”, rather instead highlighting the importance that the swelling is “persistent”. Our previous work looking at TB in meerkats suggests that lymphadenopathy is the best clinical indicator of disease (see [10]). A sentence has been added (“Animals were also visually checked for swollen lymph nodes as an indicator of likely disease”) to lines 137-138 regarding observation of disease.

Line 218: Wrong reference? Reference 12 again?

Response: This is indeed the wrong reference and it has been amended to reference 12. Thank you.

Table 2: headings look a bit messy. Suggest all columns headings are left justified with the CI on the line below. Same could be applied to Table 4 Also does DPP VetTB need a tradename symbol here?

Response: Thank you for this suggestion. We have left justified both tables 2 and 4 as suggested. We have made the appropriate adjustments to “DPP” as discussed above.

Line 245: is just ‘DPP’ OK here? Maybe the first time you use DPP (page 3) you could use the full name and bracket ‘DPP’ afterwards and then just refer to the test as ‘DPP’ from then onwards?

Response: Thank you, we have amended the use of “DPP”.

Lines 257, 326, 333,461: again inconsistence in what the DPP test is being called.

Response: Thank you, we have amended the use of “DPP”.